# Development of Shear Resistance Formula for the Y-Type Perfobond Rib Shear Connector Considering Probabilistic Characteristics

Sang-Hyo Kim, Tuvshintur Batbold, Syed Haider Ali Shah , Suro Yoon and Oneil Han *

School of Civil and Environmental Engineering, Yonsei University, Seoul 03600, Korea;
sanghyo@yonsei.ac.kr (S.-H.K.); tuvshu94@yonsei.ac.kr (T.B.); syedhaiderali@yonsei.ac.kr (S.H.A.S.);
dbstnfh112@yonsei.ac.kr (S.Y.)
* Correspondence: oneilhan@yonsei.ac.kr

**Abstract:** A design shear resistance formula for Y-type perfobond rib shear connectors is proposed with the various reduction factors, which can be selected depending on the target safety level. The nominal shear resistance formula is improved based on the systematic sensitivity analysis as well as the regression fit test based on 84 push-out test results, including 15 additional push-out tests to extend the application ranges and reduce the estimation errors, compared to the formula proposed in previous studies. Some design variables are additionally included in the proposed design formula: the yield strengths of rebar and rib plate. The basic design variables in the proposed design formula are (1) number of ribs and transverse rebars, (2) concrete compressive strength, (3) rebar diameter and yield strength, and (4) rib thickness, width, height, and yield strength. The application ranges of the basic design variables are recommended for the proposed design formula. The various shear resistance reduction factors are proposed based on the probabilistic ultimate shear resistance model of Y-type perfobond rib shear connectors. The proposed procedure may be recommended to develop the design formula for shear connectors with various shapes.

**Keywords:** Y-type perfobond rib shear connector; shear resistance formula; reduction factor; target safety level; basic design variables

## 1. Introduction

Steel and concrete composite structures have various constructional, mechanical, and economic advantages that enable them to be applied to wide range of civil engineering applications. The most benefits of composite structures can be achieved through the efficient balanced design between steel and concrete members. A shear connector is the key element in composite structures, which should guarantee the composite action and transfer the shear force between two different material bodies. Due to the various types of composite structures, many different types of shear connectors such as a stud [1], channel [2], plate perfobond [3], Y-type perfobond rib [4], and composite dowel [5] have been developed to maximize the efficiency of composite action.

The evaluation of shear resistance of a shear connector is essential in the composite structure design. Many studies have developed the shear resistance formula considering various design variables. Many shear resistance formulas for a stud shear connector have been proposed considering various conditions since Ollgaard et al. [6] first suggested it. Recently, many studies have been developed considering various design variables, such as shank diameter, stud length, material properties, loading directions [7,8], and stud spacing [9]. The plate perfobond shear connector was developed first in Germany to overcome the fatigue problems of stud. Oguejiofor and Hosain [10] suggested a shear resistance formula for a plate perfobond shear connector considering strengths of steel and concrete as well as the rib shape. Ahn et al. [11], Candido-Martins et al. [12], Zheng

et al. [13], and Yoshitaka et al. [14] have considered the distance between two adjacent plate perfobond shear connectors, geometries of the perfobond rib, and material properties. He et al. [15] compared the previously proposed shear resistance formulas and suggested a new formula. Kopp et al. [16] presented a shear resistance formula for composite dowels with PZ and CL shapes, and verifications for a beam-type section.

Kim et al. [4] introduced the Y-type perfobond rib shear connector as a new type of perfobond shear connector. The Y-type perfobond rib shear connector is one of the improved perfobond types that provides better structural performances and workability in the rebar assembly. Based on the push-out tests as well as numerical investigations, a shear resistance formula of the Y-type perfobond rib shear connector has been suggested and improved with additional experimental works by changing the basic design variables, such as rib size, transverse rebar size, concrete strength, etc.

The previous shear resistance formula has been designed to represent the individual contributions, such as the end bearing resistance, transverse rebar resistance, dowel hole resistance, Y-shape rib resistance, etc. The basic design variables, such as concrete compressive strength, Y-rib size, the number of Y-ribs and rebars, the dowel hole size for transverse rebars, the number and size of concrete blocks between adjacent Y-ribs, etc., have been included [4,17–19]. Even though the previous formula provides high accuracy to predict the shear resistances, it is a little complicated and a limitation to extend the application range.

This study is planned to extend the application range of the shear resistance formula for the Y-type perfobond rib shear connector and to improve the accuracy of the shear formula. Additional experimental specimens are designed based on the previous experiments. The new shear resistance formula is developed to represent the contributions of the main design variables, such as number of ribs and rebars, concrete strength, rib size and material strength of steel plate, rebar size and material strength of rebar, etc., rather than the contributions of individual resistance actions. The additional push-out test specimens are designed to supplement previous experimental works. The probabilistic characteristics of shear resistances of Y-shape shear connectors are investigated based on the experimental results and a Monte-Carlo simulation is performed to consider the effect of uncertain concrete strength. Based on the probabilistic characteristics of the new formula, the reduction factors are suggested to be adopted in the design process.

## 2. Supplementary Push-Out Tests for Y-Type Perfobond Rib Shear Connectors

### 2.1. Details of Additional Push-Out Test Specimens

To design the additional push-out test specimens, the experimental specimens performed in the previous studies are reviewed in terms of basic design variables. As summarized in Table 1, the previous specimens are designed with the concrete strength ranging from 30 to 50 MPa. The new specimens are designed with higher strength concrete of 60 MPa. The high-strength steel plate ribs (SM490 with a minimum yield strength of 315 MPa) are adopted to match the balance with the high concrete strength. SS400 steel has a minimum yield strength of 235 MPa. The Y-shape rib size of 80 mm-width and 100 mm-height with 10 mm-thick plate is selected as the representative size (Table 2). The various transverse rebars are adopted to investigate the rebar contribution and the effect of the balance between steel rib strength and rebar strength. SD400 rebars with a nominal yield strength of 400 MPa and SD500 rebars with 500 MPa nominal yield strength are applied. Three different diameter rebars (16, 19, 22 mm) are selected. The dowel hole diameter is fixed to be 40 mm.

**Table 1.** Design variables of previous specimens [4,17–22].

| | $f_{ck}$ | 30 MPa | 40 MPa | 50 MPa | 60 MPa |
|---|---|---|---|---|---|
| **Rebar** | | **(30.4 MPa)** | **(41.7, 42.3, 43.8 MPa)** | **(50.7, 51.0, 52.9 MPa)** | **(Planned)** |
| SD400 | D16 | SS400 | SS400 | SS400 | SM490 |
| | | - | - | SM490 | - |
| | D19 | - | SS400 | - | SM490 |
| SD500 | D16 | - | - | SS400 | SM490 |
| | | - | - | SM490 | - |
| | D19 | - | - | SS400 | SM490 |
| | D22 | - | - | SS400 | SM490 |

**Table 2.** Specimen design for additional push-out tests.

| Specimen Type | No. of Specimens | Y-Type Perfobond Rib | | | | Transverse Rebar | | Concrete Design Strength |
|---|---|---|---|---|---|---|---|---|
| | | Width | Height | Thickness | Steel Grade | Diameter | Steel Grade | |
| | | (mm) | (mm) | (mm) | | (mm) | | (MPa) |
| SD400-D16 | 3 | 80 | 100 | 10 | SM490 | 16 | SD400 | 60 |
| SD400-D19 | 3 | 80 | 100 | 10 | SM490 | 19 | SD400 | 60 |
| SD500-D16 | 3 | 80 | 100 | 10 | SM490 | 16 | SD500 | 60 |
| SD500-D19 | 4 | 80 | 100 | 10 | SM490 | 19 | SD500 | 60 |
| SD500-D22 | 2 | 80 | 100 | 10 | SM490 | 22 | SD500 | 60 |

*2.2. Fabrication of Specimens and Material Properties*

In addition to the 69 specimens tested in the previous studies, 15 4-rib specimens are designed, as listed in Table 2. Among the 69 previous specimens, 39 specimens are 4-rib specimens, and there are 27 2-rib and 3 6-rib specimens. Each test specimen consisted of one pair of n-rib shear connectors. The specimens are fabricated and tested in accordance with Eurocode-4 [23]. Four-rib Y-shape shear connectors are welded to the H-shape steel beams and embedded in two concrete blocks, as shown in Figure 1. The H-beam has a cross-section of H-300 × 300 × 9 × 14 mm, and the concrete block is 600 mm-wide, 750 mm-high, and 280 mm-thick. Four transverse rebars are placed in front of four ribs on each concrete block side. To remove the bond effects, the grease is spread on all the interfaces between steel members and concrete. The detailed shapes of the push-out test specimens are shown in Figure 1. As shown in the figures, one specimen is built up with one pair of 4-rib Y-type perfobond rib shear connectors.

Concrete compressive strength tests are performed on both day 28 and the push-out test date (Table 3). All of the cylinders are cured in the same environmental condition as the push-out specimens. The concrete is provided by two remicons. Many cylinders are fabricated from each remicon and the test results show very low deviation, especially in the compressive strengths obtained on the push-out test date. The mean values on the push-out test date are 62.2 and 62.5 MPa, and COV (coefficient of variation) is less than 2%. Therefore, one representative value of 62.4 MPa is adopted in the analysis.

The standard tensile strength tests are performed with structural steel plates and reinforcements. The Korean Standard requires the minimum yield point for SM490 to be over 315 MPa and tensile strength to be over 490 MPa [24], and the minimum yield points for SD400 and SD500 must be over 400 and 500 MPa, respectively. The tensile test results are summarized in Table 4.

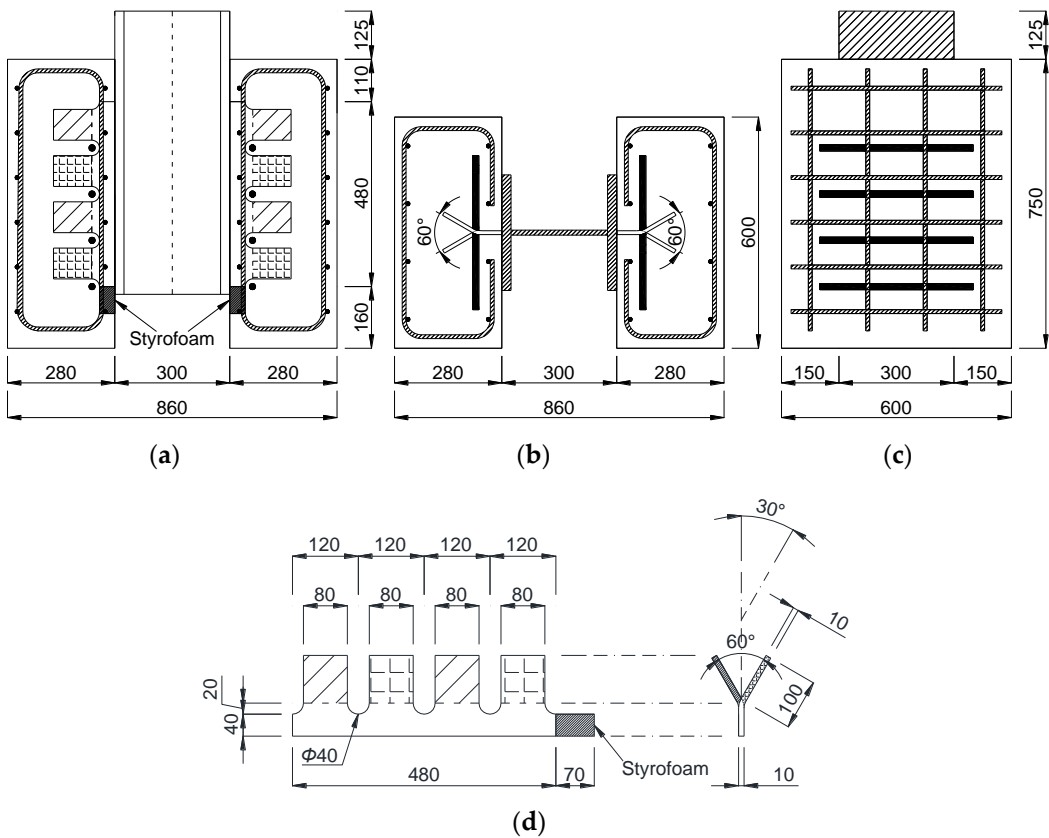

**Figure 1.** Dimensions of the push-out specimen with the Y-type perfobond rib shear connector: (**a**) Front view, (**b**) plan view, (**c**) side view. (**d**) Details of the Y-type perfobond rib shear connector.

**Table 3.** Concrete compressive strength.

| Type | Day 28 | | Push-Out Test Date | |
|---|---|---|---|---|
| | **Remicon #1** | **Remicon #2** | **Remicon #1** | **Remicon #2** |
| Experimental data (MPa) | 62.0, 61.4, 59.7, 61.9, 58.0, 61.5 | 65.7, 65.6, 63.3, 59.5, 61.9, 64.5, 60.6 | 63.5, 58.7, 62.8, 63.0, 61.0, 66.1, 63.1, 61.9, 60.8, 62.0 | 62.8, 64.6, 63.1, 63.5, 63.8, 63.1, 61.5, 61.8, 61.2, 59.8, 62.3, 65.0, 62.6 |
| Mean | 60.8 | 62.9 | 62.2 | 62.5 |
| COV | 0.013 | 0.026 | 0.015 | 0.019 |
| Average strength (MPa) | 61.8 | | 62.4 | |

**Table 4.** Rebar and steel plate tensile strength tests.

| Type | No. of Specimens | Yield Strength (MPa) | | Tensile Strength (MPa) | | Note |
|---|---|---|---|---|---|---|
| | | **Test Data** | **Average** | **Test Data** | **Average** | |
| SD400, D16 | 3 | 461, 461, 468 | 463.3 | 585, 587, 584 | 585.3 | Yield strength ≥ 400 MPa |
| SD400, D19 | 3 | 482, 479, 480 | 480.3 | 595, 594, 591 | 593.3 | |
| SD500, D16 | 3 | 534, 538, 551 | 541.0 | 659, 668, 676 | 667.7 | Yield strength ≥ 500 MPa |
| SD500, D19 | 3 | 550, 547, 543 | 546.7 | 674, 673, 670 | 672.3 | |
| SD500, D22 | 3 | 553, 549, 552 | 551.3 | 676, 672, 674 | 674.0 | |
| SM490 | 3 | 408, 406, 407 | 407.0 | 546, 550, 552 | 549.3 | Yield strength ≥ 315 MPa |

## 2.3. Push-Out Test Procedure

The push-out tests are performed following the same procedures suggested in Eurocode-4 [23] and adopted in a previous study [4]. A universal testing machine (UTM) with a capacity of 3000 kN was used for loading. The loading rate with a displacement control system was set to be 0.05 mm/s, and the push-out specimens were prevented from failing within 15 min. The push-out test was terminated when the load fell down 20% below the peak. Four 100 mm LVDTs (Linear Variable Differential Transducer) were installed at the middle of the specimen, and measured the relative slips at four points. Figure 2 shows the detailed push-out specimen and the actual test set-up of a push-out test.

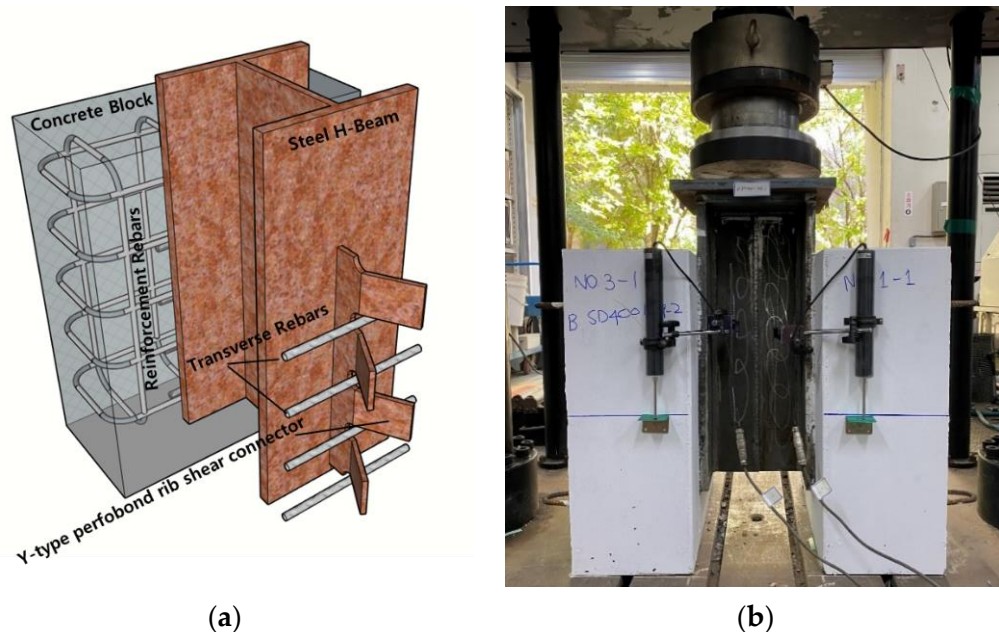

(**a**) (**b**)

**Figure 2.** Test set-up: (**a**) Concept of test specimen with the Y-type perfobond rib shear connector (4 ribs). (**b**) Actual test set-up.

## 3. New Shear Resistance Formula for the Y-Type Perfobond Rib Shear Connector

### 3.1. Current Shear Resistance Formula

The total number of the push-out test results is 84, as summarized in Table 5: 69 results from the previous studies and 15 new push-out test results from this study. Figure 3 demonstrates the load-slip curves of 5 different type specimens (4R-11, 4R-12, 4R-15, 4R-16, and 4R-18 in Table 5). The load-slip curves in Figure 3 represent the averages of 3 push-out tests in each specimen type, respectively.

Each push-out test result is the shear resistance of one pair of n-multi Y-type perfobond rib shear connectors. Therefore, 84 push-out test results are obtained with 168 n-multi Y-type perfobond rib shear connectors. Among them, 54 results are from 4-rib specimens: 39 from previous studies and 15 from this study, and 27 results are produced from 2-rib specimens. There are 3 push-out test results from 6-rib specimens. The 2-rib specimen push-out tests have mainly been performed to investigate the effect of rib shape (width, height, thickness, etc.).

**Table 5.** Shear resistance data from push-out tests and numerical evaluations.

| Test No. | | $n^{1,2)}$ | $t^{1)}$ | $w^{1)}$ | $h^{1)}$ | $d_r^{2)}$ | $f_y^{1)}$ | $f_{yr}^{2)}$ | $f_{ck}^{3)}$ | $P_u$ | Average | Reference |
|---|---|---|---|---|---|---|---|---|---|---|---|---|
| | | (ribs) | (mm) | (mm) | (mm) | (mm) | (MPa) | (MPa) | (MPa) | (kN) | (kN) | |
| 4R-1 | −1 | 4 | 10 | 80 | 100 | 16 | 235 | 400 | 30.4 | 1687.4 | 1671.9 | [4] |
| | −2 | | | | | | | | | 1636.8 | | |
| | −3 | | | | | | | | | 1691.3 | | |
| 4R-2 | −1 | 4 | 10 | 80 | 100 | 16 | 235 | 400 | 41.7 | 1821.0 | 1788.8 | [18] |
| | −2 | | | | | | | | | 1746.7 | | |
| | −3 | | | | | | | | | 1798.7 | | |
| 4R-3 | −1 | 4 | 12 | 80 | 100 | 16 | 235 | 400 | 41.7 | 2003.1 | 1972.9 | [18] |
| | −2 | | | | | | | | | 1935.3 | | |
| | −3 | | | | | | | | | 1980.4 | | |
| 4R-4 | −1 | 4 | 10 | 80 | 100 | 16 | 235 | 400 | 42.2 | 1811.1 | 1803.3 | [4] |
| | −2 | | | | | | | | | 1789.1 | | |
| | −3 | | | | | | | | | 1809.8 | | |
| 4R-5 | −1 | 4 | 10 | 80 | 100 | 16 | 235 | 400 | 43.8 | 1640.6 | 1728.6 | [19] |
| | −2 | | | | | | | | | 1746.8 | | |
| | −3 | | | | | | | | | 1798.6 | | |
| 4R-6 | −1 | 4 | 10 | 80 | 100 | 16 | 235 | 400 | 51.0 | 1949.0 | 1925.4 | [18] |
| | −2 | | | | | | | | | 1923.7 | | |
| | −3 | | | | | | | | | 1903.4 | | |
| 4R-7 | −1 | 4 | 10 | 80 | 100 | 16 | 235 | 400 | 50.7 | 2027.9 | 1948.1 | [20] |
| | −2 | | | | | | | | | 1903.9 | | |
| | −3 | | | | | | | | | 1912.5 | | |
| 4R-8 | −1 | 4 | 10 | 80 | 100 | 16 | 235 | 500 | 50.7 | 1969.0 | 1969.1 | [21] |
| | −2 | | | | | | | | | 2031.3 | | |
| | −3 | | | | | | | | | 1906.9 | | |
| 4R-9 | −1 | 4 | 10 | 80 | 100 | 16 | 315 | 400 | 52.9 | 2016.8 | 2115.8 | [22] |
| | −2 | | | | | | | | | 2137.5 | | |
| | −3 | | | | | | | | | 2193.0 | | |
| 4R-10 | −1 | 4 | 10 | 80 | 100 | 16 | 315 | 500 | 52.9 | 2107.5 | 2176.8 | [22] |
| | −2 | | | | | | | | | 2271.7 | | |
| | −3 | | | | | | | | | 2151.2 | | |
| 4R-11 | −1 | 4 | 10 | 80 | 100 | 16 | 315 | 400 | 62.4 | 2240.3 | 2228.5 | SD400-D16 |
| | −2 | | | | | | | | | 2234.3 | | (this study) |
| | −3 | | | | | | | | | 2210.9 | | |
| 4R-12 | −1 | 4 | 10 | 80 | 100 | 16 | 315 | 500 | 62.4 | 2321.3 | 2267.1 | SD500-D16 |
| | −2 | | | | | | | | | 2240.3 | | (this study) |
| | −3 | | | | | | | | | 2239.7 | | |
| 4R-13 | −1 | 4 | 10 | 80 | 100 | 19 | 235 | 400 | 43.8 | 2111.7 | 2011.1 | [17] |
| | −2 | | | | | | | | | 1903.9 | | |
| | −3 | | | | | | | | | 2017.7 | | |
| 4R-14 | −1 | 4 | 10 | 80 | 100 | 19 | 235 | 500 | 50.7 | 2124.4 | 2075.5 | [21] |
| | −2 | | | | | | | | | 2016.0 | | |
| | −3 | | | | | | | | | 2086.2 | | |
| 4R-15 | −1 | 4 | 10 | 80 | 100 | 19 | 315 | 400 | 62.4 | 2458.0 | 2470.1 | SD400-D19 |
| | −2 | | | | | | | | | 2456.1 | | (this study) |
| | −3 | | | | | | | | | 2496.3 | | |
| 4R-16 | −1 | 4 | 10 | 80 | 100 | 19 | 315 | 500 | 62.4 | 2493.2 | 2477.4 | SD500-D19 |
| | −2 | | | | | | | | | 2547.0 | | (this study) |
| | −3 | | | | | | | | | 2375.7 | | |
| | −4 | | | | | | | | | 2493.7 | | |
| 4R-17 | −1 | 4 | 10 | 80 | 100 | 22 | 235 | 500 | 50.7 | 2214.0 | 2174.9 | [21] |
| | −2 | | | | | | | | | 2211.9 | | |
| | −3 | | | | | | | | | 2098.9 | | |
| 4R-18 | −1 | 4 | 10 | 80 | 100 | 22 | 315 | 500 | 62.4 | 2614.7 | 2635.6 | SD500-D22 |
| | −2 | | | | | | | | | 2656.5 | | (this study) |

**Table 5.** *Cont.*

| Test No. | | $n^{1,2)}$ | $t^{1)}$ | $w^{1)}$ | $h^{1)}$ | $d_r^{2)}$ | $f_y^{1)}$ | $f_{yr}^{2)}$ | $f_{ck}^{3)}$ | $P_u$ | Average | Reference |
|---|---|---|---|---|---|---|---|---|---|---|---|---|
| | | (ribs) | (mm) | (mm) | (mm) | (mm) | (MPa) | (MPa) | (MPa) | (kN) | (kN) | |
| 2R-1 | −1 | 2 | 10 | 80 | 80 | 16 | 235 | 400 | 43.8 | 947.6 | 941.8 | [19] |
| | −2 | | | | | | | | | 959.2 | | |
| | −3 | | | | | | | | | 918.6 | | |
| 2R-2 | −1 | 2 | 10 | 80 | 100 | 16 | 235 | 400 | 43.8 | 991.8 | 1010.5 | [19] |
| | −2 | | | | | | | | | 1018.8 | | |
| | −3 | | | | | | | | | 1020.8 | | |
| 2R-3 | −1 | 2 | 10 | 80 | 120 | 16 | 235 | 400 | 43.8 | 1052.0 | 1041.6 | [19] |
| | −2 | | | | | | | | | 1031.8 | | |
| | −3 | | | | | | | | | 1040.8 | | |
| 2R-4 | −1 | 2 | 10 | 100 | 100 | 16 | 235 | 400 | 43.8 | 1123.2 | 1156.3 | [19] |
| | −2 | | | | | | | | | 1191.2 | | |
| | −3 | | | | | | | | | 1154.6 | | |
| 2R-5 | −1 | 2 | 10 | 120 | 100 | 16 | 235 | 400 | 43.8 | 1314.6 | 1288.1 | [19] |
| | −2 | | | | | | | | | 1254.8 | | |
| | −3 | | | | | | | | | 1294.8 | | |
| 2R-6 | −1 | 2 | 10 | 140 | 100 | 16 | 235 | 400 | 43.8 | 1343.4 | 1375.5 | [19] |
| | −2 | | | | | | | | | 1437.4 | | |
| | −3 | | | | | | | | | 1345.6 | | |
| 2R-7 | −1 | 2 | 10 | 100 | 120 | 16 | 235 | 400 | 43.8 | 1228.4 | 1217.8 | [17] |
| | −2 | | | | | | | | | 1174.8 | | |
| | −3 | | | | | | | | | 1250.2 | | |
| 2R-8 | −1 | 2 | 10 | 120 | 120 | 16 | 235 | 400 | 43.8 | 1254.4 | 1312.2 | [17] |
| | −2 | | | | | | | | | 1343.8 | | |
| | −3 | | | | | | | | | 1338.4 | | |
| 2R-9 | −1 | 2 | 10 | 140 | 120 | 16 | 235 | 400 | 43.8 | 1346.7 | 1372.3 | [17] |
| | −2 | | | | | | | | | 1407.0 | | |
| | −3 | | | | | | | | | 1363.3 | | |
| 6R-1 | −1 | 6 | 10 | 80 | 100 | 16 | 235 | 400 | 43.8 | 2478.9 | 2366.9 | [17] |
| | −2 | | | | | | | | | 2263.9 | | |
| | −3 | | | | | | | | | 2358.0 | | |
| 2R-FEA | | 2 | 10 | 80 | 100 | 16 | 235 | 400 | 40 | 1185.4 | FEM model | [17] |
| 4R-FEA | | 4 | 10 | 80 | 100 | 16 | 235 | 400 | 40 | 1805.7 | FEM model | [17] |
| 6R-FEA | | 6 | 10 | 80 | 100 | 16 | 235 | 400 | 40 | 2352.0 | FEM model | [17] |
| 8R-FEA | | 8 | 10 | 80 | 100 | 16 | 235 | 400 | 40 | 2884.4 | FEM model | [17] |
| 10R-FEA | | 10 | 10 | 80 | 100 | 16 | 235 | 400 | 40 | 3411.0 | FEM model | [17] |

[1] Y-type perfobond rib: width ($w$), height ($h$), thickness ($t$), yield strength ($f_y$), number of ribs ($n$). [2] Transverse rebar: number of transverse rebar ($n$), diameter of transverse rebar ($d_r$), yield strength of rebar ($f_{yr}$). [3] Concrete: compressive strength on the push-out test date ($f_{ck}$).

Kim et al. have improved the shear resistance formula as various design variables were considered in previous studies [4,17–19]. Equation (1) is the latest formula [17], which considers various design variables such as concrete compressive strength, the number of Y-ribs, shapes of a Y-rib, and diameters of transverse rebar:

$$Q_n = 11,500 f_{ck}^{0.3} S_{rib} + R_n f_{ck}^{0.3}\left(700 n A_{tr}^{0.75} + 2,600(n-2)S_{rib}\right) \tag{1}$$

where, the $Q_n$ (N) represents the shear resistance of the n-rib Y-type perfobond rib shear connector, $S_{rib} = t(w/80)^{0.95}(h'/120)^{0.95}(d/40)^{0.3}$ is the rib shape factor, $h' = h + 0.5d$, $R_n = 1/(n-1)^{0.12}$ is the reduction factor, $w$ (mm) is Y-rib's width, $h$ (mm) is Y-rib's height, $t$ (mm) is Y-rib's thickness, $f_{ck}$ (MPa) is concrete strength, $d$ is a dowel hole diameter, $A_{tr}$

(mm$^2$) is a cross-sectional area of a transverse rebar, and $n$ is the number of Y-ribs (or transverse rebars).

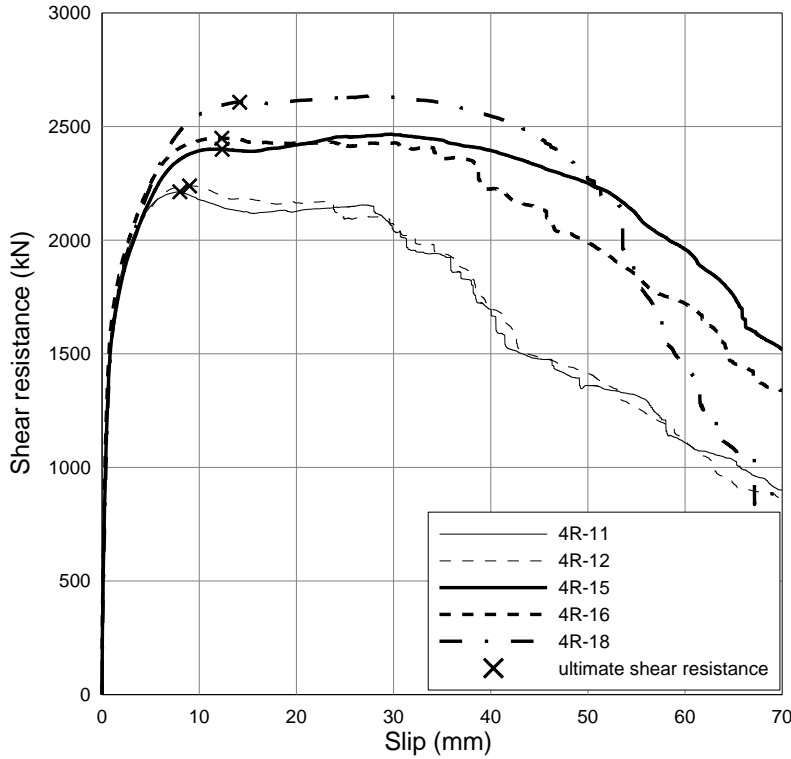

**Figure 3.** Load-slip curves of Y-type perfobond rib shear connectors (this study).

Equation (1) is designed to represent 3 main contributions, that is, the first term is for the end-bearing resistance, the second term is for the resistance by the transverse rebars, and the third term is for the resistance by Y-ribs. Because the first term provides the end bearing resistance due to the first one pair of Y-ribs located at the fronts of multi-ribs in the direction of slip, the third term contains "−2" to reduce the total number of Y-ribs. The end bearing effect may not be significant as the total number of Y-ribs increases, and it can be included in the third term. In addition, the current formula does not consider the influences of steel grades of a Y-rib and transverse rebar.

Figure 4 shows the estimation errors of Equation (1) compared to the experimental results for 2-rib, 4-rib, and 6-rib specimens. The experimental results from 2-rib specimens are found to be overestimated by the current resistance formula. It is due to the non-symmetric and unstable shape effect of only one rib in each side of Y-type shear connectors. This effect can be reduced with the increasing number of ribs. The statistical characteristics are listed in Table 6. The current resistance formula provides quite accurate shear resistance estimations for Y-type perfobond rib shear connectors with 4–6 ribs, even though the deviation is slightly high, with about 7%.

**Table 6.** Statistical characteristics of estimations by Equation (1).

| Data | No. of Data | Mean | Standard Deviation | Coefficient of Variation (COV) |
|---|---|---|---|---|
| Total | 84 | 0.972 | 0.106 (min 0.777–max 1.189) | 0.109 |
| 2 ribs | 27 | 0.844 | 0.032 (min 0.777–max 0.902) | 0.038 |
| 4 ribs | 54 | 1.034 | 0.069 (min 0.862–max 1.189) | 0.067 |
| 6 ribs | 3 | 0.992 | (min 0.949–max 1.039) | - |

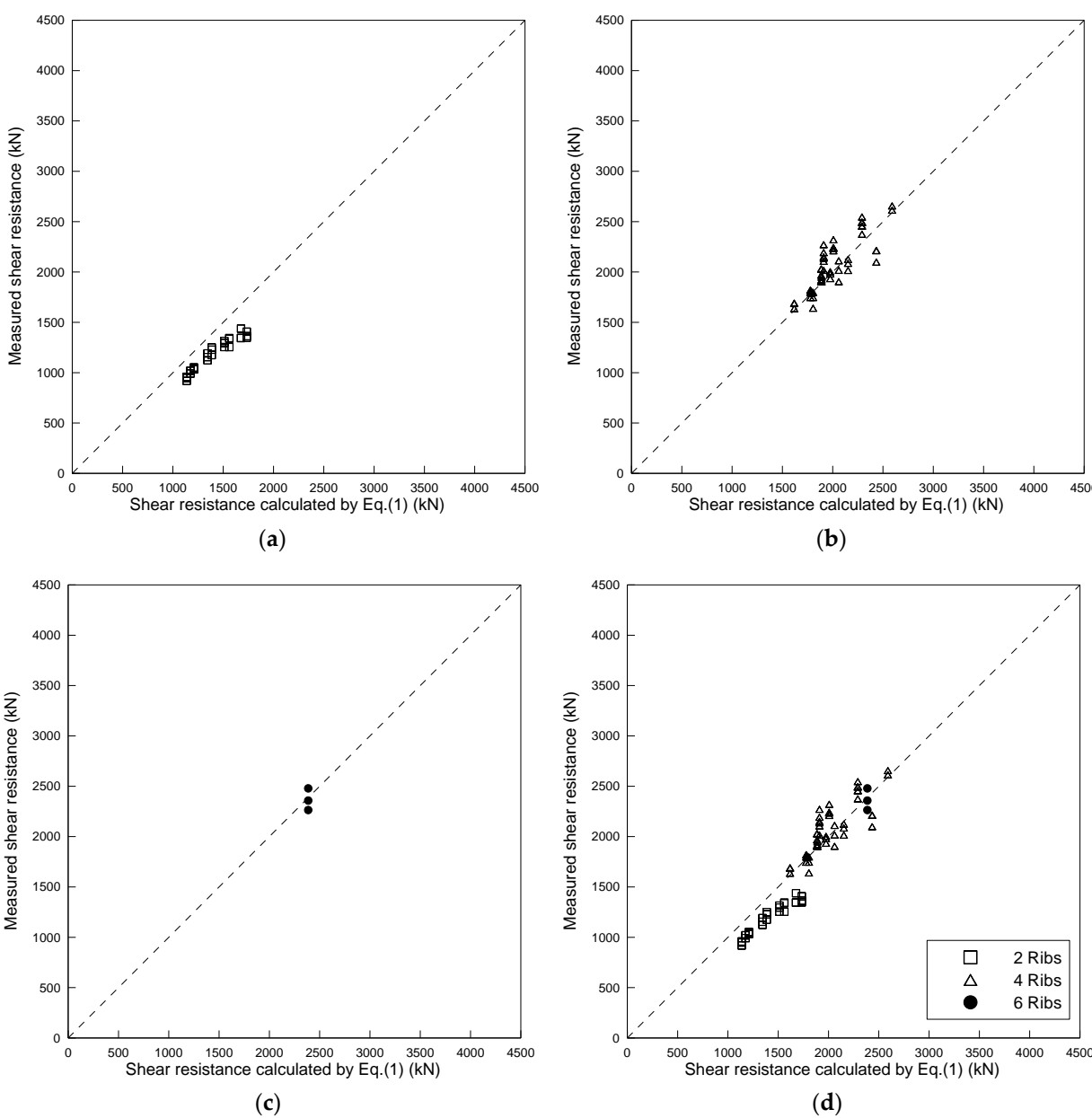

**Figure 4.** Comparison of experimental and calculated data (Equation (1)): (**a**) 2 ribs, (**b**) 4 ribs, (**c**) 6 ribs. (**d**) Total data.

### 3.2. Regression Analysis for the New Shear Resistance Formula

A new shear resistance formula is designed to extend the application range of the shear resistance formula for the Y-type perfobond rib shear connector and to improve the accuracy of shear resistance estimation, especially for multi-rib shear connectors with more than 4 ribs and high-strength concrete. The new shear resistance formula is developed to represent the contributions of main design variables, such as number of ribs and rebars, concrete strength, rib size and material strength of steel plate, rebar size and material strength of rebar, etc., rather than the contributions of individual resistance actions, such as the end bearing, dowel effect of rib, and rebar resistance, etc. A new shear resistance formula is designed to have the basic format proposed in Equation (2), in which the first term outside of the parenthesis represents the effect of multiple number of ribs. The first term in the parenthesis is the resistance due to the transverse rebar and the second term is the contribution due to Y-type rib. The last term outside of the parenthesis is the effect of the concrete strength. Therefore, the new proposed resistance formula consists of only

2 terms, that is, rebar term and rib term, and may be simpler than the current 3-term formula (Equation (1)).

$$Q_n = n^{\alpha_1}\left(\beta_1 d_r^{\alpha_2} f_{yr}^{\alpha_3} + \beta_2 f_y^{\alpha_4}\left(\frac{t}{10}\right)^{\alpha_5}\left(\frac{w}{80}\right)^{\alpha_6}\left(\frac{h}{100}\right)^{\alpha_7}\right)f_{ck}^{\alpha_8} \qquad (2)$$

where, the $Q_n$ (N) represents the nominal shear resistance of multiple Y-type perfobond rib shear connectors, $n$ is the individual number of ribs and transverse rebars, $d_r$ (mm) is the diameter of a transverse rebar, $f_{yr}$ (MPa) is the transverse rebar's yield strength, $f_y$ (MPa) is the steel plate (Y-rib) yield strength, $t$ (mm) is the steel plate (Y-rib) thickness, $w$ (mm) is the Y-rib width, $h$ (mm) is the Y-rib height, and $f_{ck}$ (MPa) is the concrete strength. All the coefficients and indices, beta and alpha values are obtained through the regression fit analysis with the experimental results. The proposed formula is derived to have the application ranges for design variables adopted in Equation (2), as summarized in Table 7, which are selected based on the experimental specimens as well as numerical specimens in Table 5.

**Table 7.** Application ranges for design variables.

| Design Variables | | Unit | Design Range |
|---|---|---|---|
| Number of Y-ribs and transverse rebars | $n$ | N/A | 4–10 |
| Concrete compressive strength | $f_{ck}$ | MPa | 30–60 |
| Diameter of transverse rebar | $d_r$ | mm | 16, 19, 22 |
| Yield strength of transverse rebar | $f_{yr}$ | MPa | SD400 ($f_{yr}$ = 400) SD500 ($f_{yr}$ = 500) |
| Y-rib thickness | $t$ | mm | 10–12 |
| Y-rib width | $w$ | mm | 80–120 |
| Y-rib height | $h$ | mm | 80–120 |
| Yield strength of structural steel | $f_y$ | MPa | SS400 ($f_y$ = 235) SM490 ($f_y$ = 315) |

### 3.2.1. Investigation of Index Value for Concrete Strength Term: $f_{ck}$

In advance of the regression fit test with the whole equation (Equation (2)) to select the proper index values (alpha values) adopted in Equation (2), the trial index values are first selected based on the individual sensitivity analysis with the basic design variables.

To investigate the trial index value for the concrete compressive strength ($f_{ck}$), 4 specimen groups are selected from the results in Table 5 and summarized in Table 8. The sensitivity analysis is performed with the (1) 30.4 MPa group to 42.6 MPa group, (2) 42.6 MPa group to 50.9 MPa group, (3) 52.9 MPa group to 62.4 MPa group, and (4) another 52.9 MPa group to another 62.4 MPa group with different rebar grade (SD400 and SD500 groups). All specimens in each group have the same design variables except the concrete strength. Based on the results in Table 8, the index of 0.3 is proposed for the regression fit test with the experimental results.

**Table 8.** Index values for the variable of $f_{ck}$.

| Group 1 | | Group 2 | | Ratio | | Index (G = logF/logE) | Test No. |
|---|---|---|---|---|---|---|---|
| $f_{ck}$(A) | $P_u$(B) | $f_{ck}$(C) | $P_u$(D) | $f_{ck}$ (E = C/A) | $P_u$ (F = D/B) | | |
| 30.4 | 1671.9 | 42.6 | 1773.6 | 1.400 | 1.061 | 0.175 | 4R-1, 2, 4, 5 |
| 42.6 | 1773.6 | 50.9 | 1936.7 | 1.195 | 1.092 | 0.495 | 4R-2, 4, 5, 6, 7 |
| 52.9 | 2115.8 | 62.4 | 2228.5 | 1.180 | 1.053 | 0.314 | 4R-9, 11 |
| 52.9 | 2176.8 | 62.4 | 2267.1 | 1.180 | 1.041 | 0.246 | 4R-10, 12 |
| | | Average | | | | 0.308 | |

### 3.2.2. Investigation of Index Values for Rebar: $f_{yr}$ and $d_r$

To investigate the trial index value of $f_{yr}$, 4 specimen groups are selected as in Table 9 and the trial index of 0.2 is selected based on the sensitivity analysis, in which one result of 0.029 is excluded. All specimens in each group have the same design variables except the yield strength of rebar. In Table 5, one specimen result of test group 4R-16 shows a very low ultimate shear strength of 2375.7 kN. If this result is excluded, the mean value of the test group increases to 2511.3 kN and the ratio of $P_u$ for the test group in Table 9 will be 1.017 rather than 1.003. There are many specimen groups related to the rebar diameters, as summarized in Table 10. Based on the sensitivity analysis, the index of 1.0 is selected for $d_r$. All specimens in each group have the same design variables except the rebar diameter.

**Table 9.** Index value for the variable of $f_{yr}$.

| Group 1 | | Group 2 | | Ratio | | Index (G = 2.2logF/logE) | Test No. |
|---|---|---|---|---|---|---|---|
| $f_{yr}$ (A) | $P_u$ (B) | $f_{yr}$ (C) | $P_u$ (D) | $f_{yr}$ (E = C/A) | $P_u$ (F = D/B) | | |
| 400 | 1936.7 | 500 | 1969.1 | 1.250 | 1.017 | 0.165 | 4R-6, 7, 8 |
| 400 | 2115.8 | 500 | 2176.8 | 1.250 | 1.029 | 0.283 | 4R-9, 10 |
| 400 | 2228.5 | 500 | 2267.1 | 1.250 | 1.017 | 0.171 | 4R-11, 12 |
| 400 | 2470.1 | 500 | 2477.4 | 1.250 | 1.003 | 0.029 * | 4R-15, 16 |
| Average | | | | | | 0.206 | |

\* excluded from calculating the average.

**Table 10.** Index value for the variable of $d_r$.

| Group 1 | | Group 2 | | Ratio | | Index (G = 2.2logF/logE) | Test No. |
|---|---|---|---|---|---|---|---|
| $d_r$(A) | $P_u$(B) | $d_r$(C) | $P_u$(D) | $d_r$ (E = C/A) | $P_u$ (F = D/B) | | |
| 16 | 1773.6 | 19 | 2011.1 | 1.188 | 1.134 | 1.625 | 4R-2, 4, 5, 13 |
| 16 | 1969.1 | 19 | 2075.5 | 1.188 | 1.054 | 0.681 | 4R-8, 14 |
| 16 | 1969.1 | 22 | 2174.9 | 1.375 | 1.105 | 0.694 | 4R-8, 17 |
| 19 | 2075.5 | 22 | 2174.9 | 1.158 | 1.048 | 0.709 | 4R-14, 17 |
| 16 | 2228.5 | 19 | 2470.1 | 1.188 | 1.108 | 1.331 | 4R-11, 15 |
| 16 | 2267.1 | 19 | 2477.4 | 1.188 | 1.093 | 1.147 | 4R-12, 16 |
| 16 | 2267.1 | 22 | 2635.6 | 1.375 | 1.163 | 1.051 | 4R-12, 18 |
| 19 | 2477.4 | 22 | 2635.6 | 1.158 | 1.064 | 0.938 | 4R-16, 18 |
| Average | | | | | | 1.022 | |

### 3.2.3. Investigation of Index Values for Ribs: $f_y$, $t$, $w$, and $h$

The sensitivity analysis results are summarized in Tables 11–14 for 4 design variables related to the steel ribs. Based on the trial index values in Tables 11–14, the index value of 0.5 is finally selected for $f_y$ through the regression fit test, the index value of 0.95 for $w$, the index value of 0.45 for $h$, and the index value of 1.0 for $t$.

**Table 11.** Index value for the variable of $f_y$.

| Group 1 | | Group 2 | | Ratio | | Index (G = 1.8logF/logE) | Test No. |
|---|---|---|---|---|---|---|---|
| $f_y$(A) | $P_u$(B) | $f_y$(C) | $P_u$(D) | $f_y$ (E = C/A) | $P_u$ (F = D/B) | | |
| 235 | 1936.7 | 315 | 2115.8 | 1.340 | 1.092 | 0.549 | 4R-6, 7, 9 |
| 235 | 1969.1 | 315 | 2176.8 | 1.340 | 1.105 | 0.622 | 4R-8, 10 |
| Average | | | | | | 0.585 | |

The increasing rate of the shear resistance with the increasing width of rib reduces beyond 120 mm, as shown in Table 12. Therefore, the application range for the rib width is suggested up to 120 mm, even though the sensitivity analysis includes the results of 140 mm-wide specimens. The proper rib size should be selected considering the balanced

design among rib height, rib width, rib thickness, and rebar diameter. The material strengths of rib and rebar need to be considered.

**Table 12.** Index value for the variable of $w$.

| Group 1 | | Group 2 | | Ratio | | Index (G = 1.8logF/logE) | Test No. |
|---|---|---|---|---|---|---|---|
| $w(\text{A})$ | $P_u(\text{B})$ | $w(\text{C})$ | $P_u(\text{D})$ | $w$ (E = C/A) | $P_u$ (F = D/B) | | |
| 80 | 1010.5 | 100 | 1156.3 | 1.250 | 1.144 | 1.099 | 2R-2, 4 |
| 80 | 1010.5 | 120 | 1288.1 | 1.500 | 1.275 | 1.088 | 2R-2, 5 |
| 80 | 1010.5 | 140 | 1375.5 | 1.750 | 1.361 | 1.002 | 2R-2, 6 |
| 80 | 1041.6 | 100 | 1217.8 | 1.250 | 1.169 | 1.274 | 2R-3, 7 |
| 80 | 1041.6 | 120 | 1312.2 | 1.500 | 1.260 | 1.036 | 2R-3, 8 |
| 80 | 1041.6 | 140 | 1372.3 | 1.750 | 1.318 | 0.896 | 2R-3, 9 |
| 100 | 1156.3 | 120 | 1288.1 | 1.200 | 1.114 | 1.076 | 2R-4, 5 |
| 100 | 1156.3 | 140 | 1375.5 | 1.400 | 1.190 | 0.938 | 2R-4, 6 |
| 120 | 1288.1 | 140 | 1375.5 | 1.167 | 1.068 | 0.774 | 2R-5, 6 |
| 100 | 1217.8 | 120 | 1312.2 | 1.200 | 1.078 | 0.744 | 2R-7, 8 |
| 100 | 1217.8 | 140 | 1372.3 | 1.400 | 1.127 | 0.646 | 2R-7, 9 |
| Average | | | | | | 0.961 | |

**Table 13.** Index value for the variable of $h$.

| Group 1 | | Group 2 | | Ratio | | Index (G = 1.8logF/logE) | Test No. |
|---|---|---|---|---|---|---|---|
| $h(\text{A})$ | $P_u(\text{B})$ | $h(\text{C})$ | $P_u(\text{D})$ | $h$ (E = C/A) | $P_u$ (F = D/B) | | |
| 80 | 941.8 | 100 | 1010.5 | 1.250 | 1.073 | 0.574 | 2R-1, 2 |
| 80 | 941.8 | 120 | 1041.6 | 1.500 | 1.106 | 0.452 | 2R-1, 3 |
| 100 | 1010.5 | 120 | 1041.6 | 1.200 | 1.031 | 0.302 | 2R-2, 3 |
| 100 | 1156.3 | 120 | 1217.8 | 1.200 | 1.053 | 0.516 | 2R-4, 7 |
| Average | | | | | | 0.461 | |

**Table 14.** Index value for the variable of $t$.

| Group 1 | | Group 2 | | Ratio | | Index (G = 1.8logF/logE) | Test No. |
|---|---|---|---|---|---|---|---|
| $t(\text{A})$ | $P_u$ (B) | $t(\text{C})$ | $P_u$ (D) | $t$ (E = C/A) | $P_u$ (F = D/B) | | |
| 10 | 1773.6 | 12 | 1972.9 | 1.200 | 1.112 | 1.062 | 4R-2, 3, 4, 5 |
| Average | | | | | | 1.062 | |

### 3.2.4. Investigation of Index Value for Number of Ribs and Rebars: $n$

The shear resistance of multiple Y-type perfobond rib shear connectors may not increase linearly with the number of ribs. It is found that the contribution of the additional rib decreases and then converges to a certain resistance [17]. To investigate the sensitivity of the shear resistance with respect to the number of ribs and rebars, the numerical evaluation results are employed rather than the push-out test results, because the experimental results show the unstable resistances for the specimens with small number of ribs and the total number of ribs in the experimental specimens are limited due to the limit of the loading capacity. The numerical evaluations are adopted from the previous study [17] and the numerical results have been verified with the various push-out test results. Based on the sensitivity analysis summarized in Table 15, with various combinations ranging from 2 ribs to 10 ribs, the trial index values between 0.67 to 0.69 are tested and 0.67 is finally selected through the regression fit test. It should be noticed that the index value of 0.67 provides a conservative estimation (underestimation) compared to the formula with a higher index value. It will be discussed in Section 3.2.5.

**Table 15.** Index value for the variable of *n*.

| Group 1 | | Group 2 | | Ratio | | Index (G = logF/logE) | Test No. |
|---|---|---|---|---|---|---|---|
| *n*(A) | $P_u$(B) | *n*(C) | $P_u$(D) | *n* (E = C/A) | $P_u$ (F = D/B) | | |
| 2 | 1185.4 | 4 | 1805.7 | 2.000 | 1.523 | 0.607 | 2R/4R-FEA |
| 4 | 1805.7 | 6 | 2352.0 | 1.500 | 1.303 | 0.652 | 4R/6R-FEA |
| 6 | 2352.0 | 8 | 2884.4 | 1.333 | 1.226 | 0.709 | 6R/8R-FEA |
| 8 | 2884.4 | 10 | 3411.0 | 1.250 | 1.183 | 0.751 | 8R/10R-FEA |
| 2 | 1185.4 | 6 | 2352.0 | 3.000 | 1.984 | 0.624 | 2R/6R-FEA |
| 2 | 1185.4 | 8 | 2884.4 | 4.000 | 2.433 | 0.641 | 2R/8R-FEA |
| 2 | 1185.4 | 10 | 3411.0 | 5.000 | 2.878 | 0.657 | 2R/10R-FEA |
| 4 | 1805.7 | 8 | 2884.4 | 2.000 | 1.597 | 0.676 | 4R/8R-FEA |
| 4 | 1805.7 | 10 | 3411.0 | 2.500 | 1.889 | 0.694 | 4R/10R-FEA |
| 6 | 2352.0 | 10 | 3411.0 | 1.667 | 1.450 | 0.728 | 6R/10R-FEA |
| Average | | | | | | 0.674 | |

3.2.5. New Shear Resistance Formula

The two coefficients, $\beta_1$ and $\beta_2$, for the two terms in the new shear resistance formula (Equation (2)) represent the contributions related to rebar and rib, and it is found that the contributions are about 45% and 55% of a total of shear resistance, respectively. The optimal values of $\beta_1$ and $\beta_2$ are searched with the fixed contribution ratio between two terms as well as the index values of alpha selected through the repeated regression fit tests. The following new nominal shear resistance formula for a multiple Y-type perfobond rib shear connector, Equation (3), is proposed:

$$Q_n = n^{0.67} \left( 970 d_r f_{yr}^{0.2} + 4240 \sqrt{f_y} \left( \frac{t}{10} \right) \left( \frac{w}{80} \right)^{0.95} \left( \frac{h}{100} \right)^{0.45} \right) f_{ck}^{0.3} \qquad (3)$$

where, the $Q_n$ (N) represents the nominal shear resistance of multiple Y-type perfobond rib shear connectors, *n* is the individual number of ribs and transverse rebars, $d_r$ (mm) is the diameter of a transverse rebar, $f_{yr}$ (MPa) is the transverse rebar's yield strength, $f_y$ (MPa) is the steel plate (Y-rib) yield strength, *t* (mm) is the steel plate (Y-rib) thickness, *w* (mm) is the Y-rib width, *h* (mm) is the Y-rib height, and $f_{ck}$ (MPa) is the concrete strength. To calculate the nominal resistance, the nominal values need to be input for all design variables in Equation (3).

In order to verify the accuracy of the new formula, Figure 5 compares the shear resistances from experimental tests with the estimation results (Equation (3)). The datasets are grouped into four parts, such as 2-rib (27 data: Figure 5a), 4-rib (54 data: Figure 5b), 6-rib (3 data: Figure 5c), and total (84 data: Figure 5d). The estimation resistance is calculated with the nominal values for all design variables except the concrete compressive strength, which is adopted from the cylinder tests for the test specimens.

As summarized in Tables 16 and 17, the new shear resistance formula (Equation (3)) provides better estimations than the current formula (Equation (1)). The estimations range from 0.794 to 1.064, which is better than the current estimation ranging from 0.777 to 1.189 in Table 6. The COV is reduced from 0.109 (current formula) to 0.074 (proposed formula). Especially for 4-rib specimens, the estimation errors range from 0.895 to 1.064 with a mean of 1.000 (COV of 0.033) in the proposed formula, whereas the current formula ranges from 0.862 to 1.189 with a mean of 1.034 (COV of 0.067). All the ultimate shear resistances in Figure 5 and Table 17 are the resistances of one-pair sets of n-rib specimens. The proposed formula generally overestimates the resistances for the specimens with 2 ribs. However, the estimation errors are improved from 0.777–0.902 with mean of 0.844 to 0.794–0.914 with mean of 0.865.

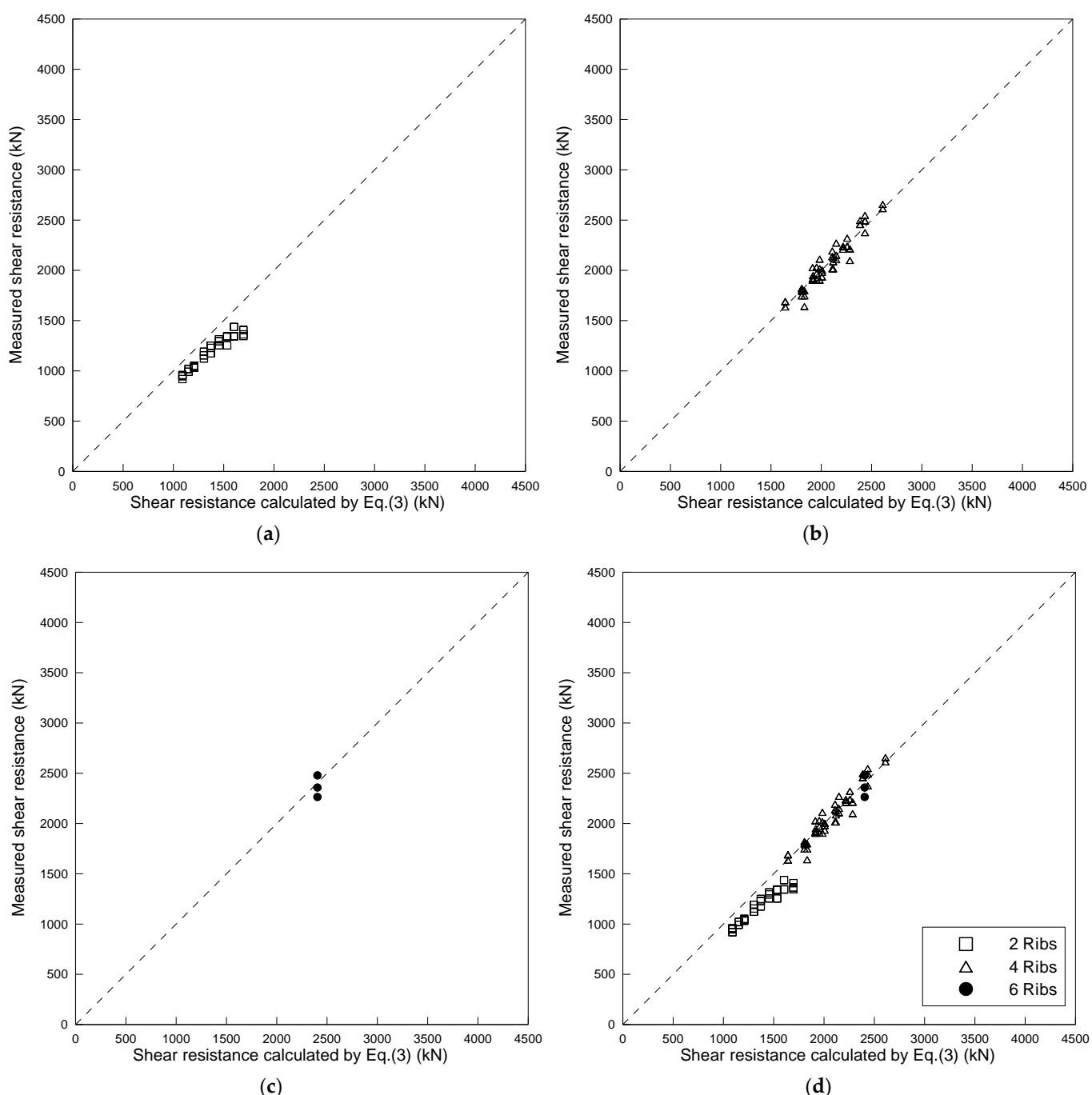

**Figure 5.** Comparison of experimental and calculated data (Equation (3)): (**a**) 2 ribs, (**b**) 4 ribs, (**c**) 6 ribs, (**d**) total data.

**Table 16.** Statistical characteristics of the proposed resistance formula (Experimental result/Estimation by Equation (3)).

| Data | No. of Data | Mean | Standard Deviation | Coefficient of Variation (COV) |
|------|------------|------|--------------------|-------------------------------|
| Total | 84 | 0.956 | 0.071 (min 0.794–max 1.064) | 0.074 |
| 2 ribs | 27 | 0.865 | 0.031 (min 0.794–max 0.914) | 0.036 |
| 4 ribs | 54 | 1.000 | 0.033 (min 0.895–max 1.064) | 0.033 |
| 6 ribs | 3 | 0.985 | (min 0.942–max 1.031) | - |

**Table 17.** Comparison of experimental resistance and estimated results (Equation (3)).

| Test No. | | Exp. $P_u$ (kN)(A) | Equation (3) Avg. (kN)(B) | $Q_n$ (kN)(C) | Ratio (A/C) | Ratio (B/C) | Test No. | | Exp. $P_u$ (kN)(A) | Equation (3) Avg. (kN)(B) | $Q_n$ (kN)(C) | Ratio (A/C) | Ratio (B/C) |
|---|---|---|---|---|---|---|---|---|---|---|---|---|---|
| 4R-1 | −1 | 1687.4 | 1671.9 | 1642.0 | 1.03 | 1.02 | 4R-15 | −1 | 2458.0 | 2470.1 | 2385.5 | 1.03 | 1.04 |
| | −2 | 1636.8 | | | 1.00 | | | −2 | 2456.1 | | | 1.03 | |
| | −3 | 1691.3 | | | 1.03 | | | −3 | 2496.3 | | | 1.05 | |
| 4R-2 | −1 | 1821.0 | 1788.8 | 1805.3 | 1.01 | 0.99 | 4R-16 | −1 | 2493.2 | 2477.4 | 2434.3 | 1.02 | 1.02 |
| | −2 | 1746.7 | | | 0.97 | | | −2 | 2547.0 | | | 1.05 | |
| | −3 | 1798.7 | | | 1.00 | | | −3 | 2375.7 | | | 0.98 | |
| 4R-3 | −1 | 2003.1 | 1972.9 | 2006.8 | 1.00 | 0.98 | | −4 | 2493.7 | | | 1.02 | |
| | −2 | 1935.3 | | | 0.96 | | 4R-17 | −1 | 2214.0 | 2174.9 | 2284.5 | 0.97 | 0.95 |
| | −3 | 1980.4 | | | 0.99 | | | −2 | 2211.9 | | | 0.97 | |
| 4R-4 | −1 | 1811.1 | 1803.3 | 1811.8 | 1.00 | 1.00 | | −3 | 2098.9 | | | 0.92 | |
| | −2 | 1789.1 | | | 0.99 | | 4R-18 | −1 | 2614.7 | 2635.6 | 2610.8 | 1.00 | 1.01 |
| | −3 | 1809.8 | | | 1.00 | | | −2 | 2656.5 | | | 1.02 | |
| 4R-5 | −1 | 1640.6 | 1728.6 | 1832.1 | 0.90 | 0.94 | 2R-1 | −1 | 947.6 | 941.8 | 1090.1 | 0.87 | 0.86 |
| | −2 | 1746.8 | | | 0.95 | | | −2 | 959.2 | | | 0.88 | |
| | −3 | 1798.6 | | | 0.98 | | | −3 | 918.6 | | | 0.84 | |
| 4R-6 | −1 | 1949.0 | 1925.4 | 1917.7 | 1.02 | 1.00 | 2R-2 | −1 | 991.8 | 1010.5 | 1151.5 | 0.86 | 0.88 |
| | −2 | 1923.7 | | | 1.00 | | | −2 | 1018.8 | | | 0.88 | |
| | −3 | 1903.4 | | | 0.99 | | | −3 | 1020.8 | | | 0.89 | |
| 4R-7 | −1 | 2027.9 | 1948.1 | 1914.3 | 1.06 | 1.02 | 2R-3 | −1 | 1052.0 | 1041.6 | 1206.4 | 0.87 | 0.86 |
| | −2 | 1903.9 | | | 0.99 | | | −2 | 1031.8 | | | 0.86 | |
| | −3 | 1912.5 | | | 1.00 | | | −3 | 1040.8 | | | 0.86 | |
| 4R-8 | −1 | 1969.0 | 1969.1 | 1952.9 | 1.01 | 1.01 | 2R-4 | −1 | 1123.2 | 1156.3 | 1303.3 | 0.86 | 0.89 |
| | −2 | 2031.3 | | | 1.04 | | | −2 | 1191.2 | | | 0.91 | |
| | −3 | 1906.9 | | | 0.98 | | | −3 | 1154.6 | | | 0.89 | |
| 4R-9 | −1 | 2016.8 | 2115.8 | 2109.6 | 0.96 | 1.00 | 2R-5 | −1 | 1314.6 | 1288.1 | 1453.5 | 0.90 | 0.89 |
| | −2 | 2137.5 | | | 1.01 | | | −2 | 1254.8 | | | 0.86 | |
| | −3 | 2193.0 | | | 1.04 | | | −3 | 1294.8 | | | 0.89 | |
| 4R-10 | −1 | 2107.5 | 2176.8 | 2148.7 | 0.98 | 1.01 | 2R-6 | −1 | 1343.4 | 1375.5 | 1602.5 | 0.84 | 0.86 |
| | −2 | 2271.7 | | | 1.06 | | | −2 | 1437.4 | | | 0.90 | |
| | −3 | 2151.2 | | | 1.00 | | | −3 | 1345.6 | | | 0.84 | |
| 4R-11 | −1 | 2240.3 | 2228.5 | 2216.8 | 1.01 | 1.01 | 2R-7 | −1 | 1228.4 | 1217.8 | 1371.2 | 0.90 | 0.89 |
| | −2 | 2234.3 | | | 1.01 | | | −2 | 1174.8 | | | 0.86 | |
| | −3 | 2210.9 | | | 1.00 | | | −3 | 1250.2 | | | 0.91 | |
| 4R-12 | −1 | 2321.3 | 2267.1 | 2257.8 | 1.03 | 1.00 | 2R-8 | −1 | 1254.4 | 1312.2 | 1534.3 | 0.82 | 0.86 |
| | −2 | 2240.3 | | | 0.99 | | | −2 | 1343.8 | | | 0.88 | |
| | −3 | 2239.7 | | | 0.99 | | | −3 | 1338.4 | | | 0.87 | |
| 4R-13 | −1 | 2111.7 | 2011.1 | 1983.9 | 1.06 | 1.01 | 2R-9 | −1 | 1346.7 | 1372.3 | 1696.1 | 0.79 | 0.81 |
| | −2 | 1903.9 | | | 0.96 | | | −2 | 1407.0 | | | 0.83 | |
| | −3 | 2017.7 | | | 1.02 | | | −3 | 1363.3 | | | 0.80 | |
| 4R-14 | −1 | 2124.4 | 2075.5 | 2118.7 | 1.00 | 0.98 | 6R-1 | −1 | 2478.9 | 2366.9 | 2404.0 | 1.03 | 0.98 |
| | −2 | 2016.0 | | | 0.95 | | | −2 | 2263.9 | | | 0.94 | |
| | −3 | 2086.2 | | | 0.98 | | | −3 | 2358.0 | | | 0.98 | |

The proposed formula, Equation (3), is investigated with the numerical estimations with the different number of ribs. The proposed formula is found to slightly provide overestimations as the number of ribs increases, as shown in Figure 6 and Table 18. This trend may be modified by taking a smaller index value for *n* in Equation (3). When the index value for *n* is reduced, the underestimation errors in 2-rib and 4-rib specimens may increase, whereas the overestimation errors in 6- to 10-rib specimens may be reduced. Based on many trials to improve the estimation errors, the proposed formula is selected as the optimum estimation equation. This estimation error may be overcome by selecting a proper resistance reduction factor considering the aleatory probabilistic characteristics inherent in the shear resistances of Y-type perfobond rib shear connectors.

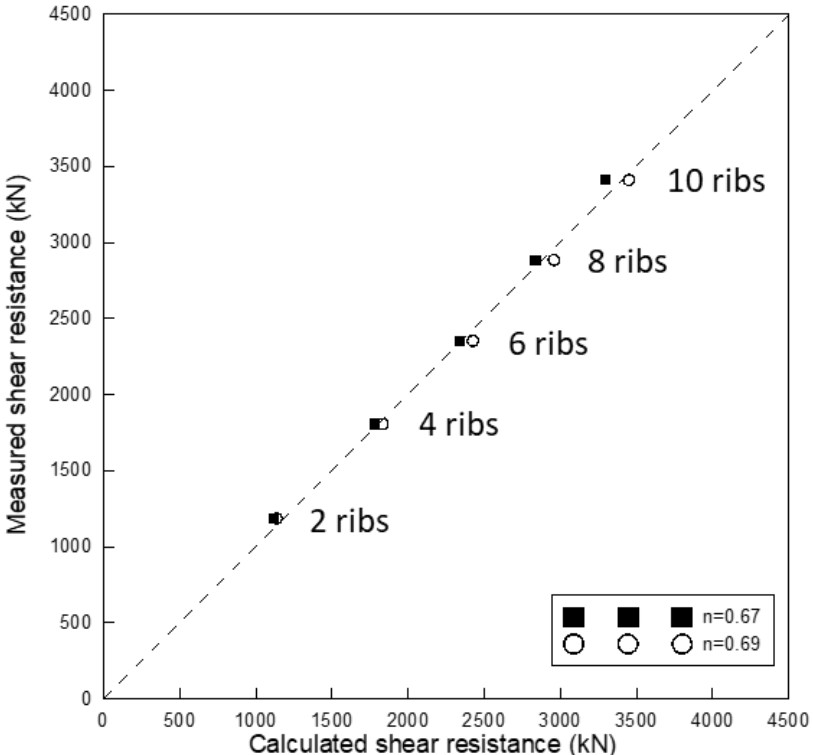

**Figure 6.** Comparison of numerical and calculated results.

**Table 18.** Biasness of the proposed formula (Equation (3)).

| Test No. | FEA $P_u$ | Equation (3) Using $n^{0.67}$ $Q_n$ | Ratio | Equation (3) Using $n^{0.69}$ $Q_n$ | Ratio |
|---|---|---|---|---|---|
| | (kN) (A) | (kN) (B) | (A/B) | (kN) (C) | (A/C) |
| 2R-FEA | 1185.4 | 1120.6 | 1.06 | 1136.2 | 1.04 |
| 4R-FEA | 1805.7 | 1782.9 | 1.01 | 1833.0 | 0.99 |
| 6R-FEA | 2352.0 | 2339.4 | 1.01 | 2424.8 | 0.97 |
| 8R-FEA | 2884.4 | 2836.7 | 1.02 | 2957.2 | 0.98 |
| 10R-FEA | 3411.0 | 3294.2 | 1.04 | 3449.4 | 0.99 |

## 4. Reduction Factor for the Y-Type Perfobond Rib Shear Resistance

### 4.1. Probabilistic Characteristics of the Shear Resistance

The probabilistic characteristics of the shear resistance of Y-type perfobond rib shear connector are investigated with the proposed resistance formula, Equation (3). The proposed formula contains many uncertainty sources from the design variable as well as the formula itself. The uncertain characteristics inherent in the proposed formula and all the design variables except the concrete strength are already included in the estimation summarized in Table 16 and Figure 5, in which the experimental concrete strengths are adopted from the cylinder tests. The probabilistic characteristics are evaluated with the experimental results of 4-rib specimens, in which the sample size is 54. The experimental results of 27 2-rib specimens and 3 6-rib specimens are excluded to guarantee the homogeneous specimens. Figure 7 shows 54 experimental results plotted on a normal probability paper and two linear regression lines. One is the regression line with all 54 data and the other is the regression line with the lower 27 data. Even though two regression lines do not show the noticeable difference in Figure 7 and Table 19, the regression results (solid line in Figure 7) obtained with the lower part of the data is adopted to provide a conservative model.

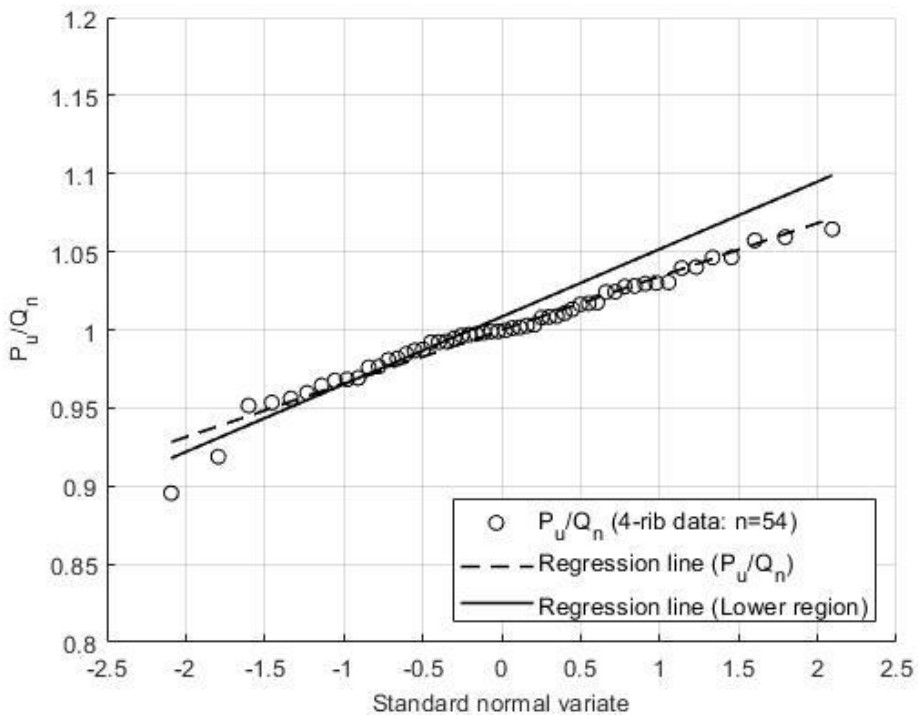

**Figure 7.** Plot of experimental results on a normal PDF paper.

**Table 19.** Probabilistic characteristics of 4-rib experimental data.

| Type | Mean | COV | No. of Samples (n) | Correlation Coefficient |
|---|---|---|---|---|
| Regression line (whole data) | 1.000 | 0.034 | 54 | 0.9816 |
| Regression line (Lower region data: s ≤ 0) | 1.008 | 0.043 | 27 | 0.9625 |

The probabilistic characteristics of the shear resistance including the uncertainty of the concrete strength are simulated combining two uncertainty sources, one from the concrete strength and another from the rest part in the proposed formula (Equation (3)). The uncertainty model for the other part is the normal PDF with mean of 1.008 and COV of 0.043. The probabilistic model for the concrete strength is adopted from the previous study [25]. The basic models are summarized in Table 20. The simulation is performed through the Monte-Carlo simulation procedure. The first part of Equation (3), except $f_{ck}^{0.3}$, is generated based on the normal PDF with a mean of 1.008 and a COV of 0.043, in which $f_{ck}$ is assumed to be a deterministic normalized value. The $f_{ck}^{0.3}$ part is generated based on another normal PDF with a mean of 1.120 and a COV of 0.120. Then, two parts are multiplied. The Monte-Carlo simulation results are also plotted on a normal probability paper (Figure 8) and the regression model is summarized in Table 21 with the statistical estimations. The mean and COV obtained from the linear regression on the probability paper are almost the same as those from the statistical estimation.

**Table 20.** Probabilistic models for Monte-Carlo simulation with the 4-rib case.

| Type | PDF | Mean | COV |
|---|---|---|---|
| Shear resistance (except $f_{ck}$) | Normal | 1.008 | 0.043 |
| $f_{ck}$ | Normal | 1.120 | 0.120 |

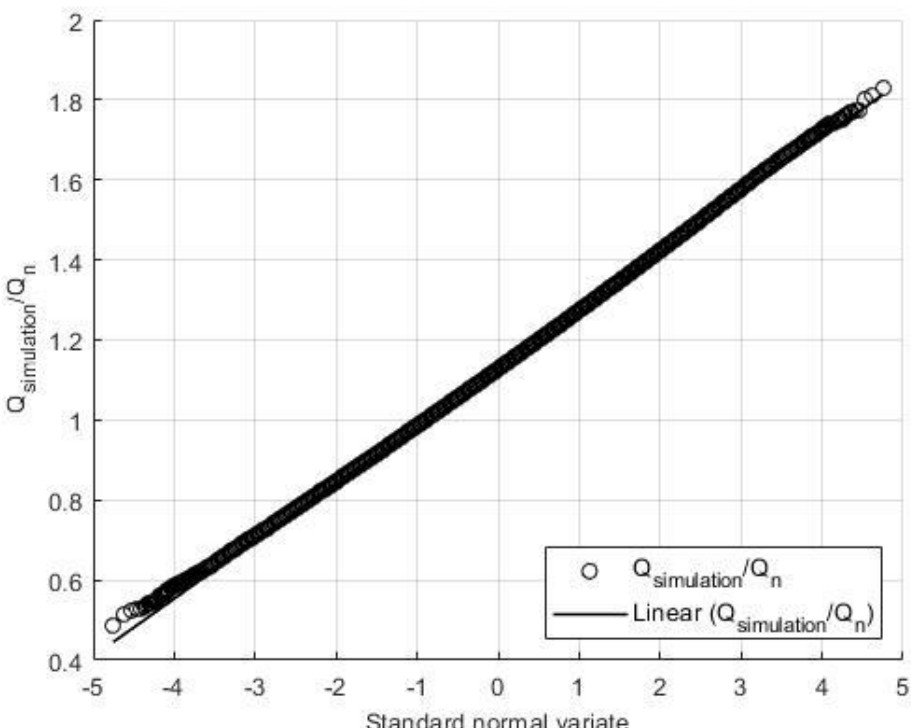

**Figure 8.** Plot of the simulated shear resistances on a normal PDF paper.

**Table 21.** Probabilistic characteristics of the shear resistance from Monte-Carlo simulation.

|  | Mean | COV | n | Correlation Coefficient |
|---|---|---|---|---|
| Sample data | 1.129 | 0.127 | 1,000,000 | - |
| Linear regression | 1.129 | 0.127 | 1,000,000 | 0.9998 |

### 4.2. Design Shear Resistance for the Y-Type Perfobond Rib Shear Connector

Based on the probabilistic models of the shear resistance of the Y-type perfobond rib shear connector (Table 22), the reduction factor ($\phi$) is derived to provide the design shear resistance depending on the target reliability level. The design shear resistance ($Q_d$) can be calculated from the reduction factor and the nominal shear resistance ($Q_n$, Equation (3)) by either Equation (4) or Equation (5):

$$Q_d = \phi Q_n \tag{4}$$

$$Q_d = \phi n^{0.67}\left(970 d_r f_{yr}^{0.2} + 4,240\sqrt{f_y}\left(\frac{t}{10}\right)\left(\frac{w}{80}\right)^{0.95}\left(\frac{h}{100}\right)^{0.45}\right) f_{ck}^{0.3} \tag{5}$$

**Table 22.** Probabilistic models of 4- and 10-rib cases.

| Case | PDF | Mean | COV |
|---|---|---|---|
| 4-rib | Normal | 1.129 | 0.127 |
| 10-rib | Normal | 1.155 | 0.127 |

The reduction factors ($\phi$) are provided in the wide range from 0.50 to 0.90, and the safety levels to be achieved are tabulated in Table 23. The worst estimation error by the proposed nominal resistance is 0.90 (experimental result/Equation (3): Test No. 4R-5-1 in Table 17), in which the estimation by Equation (3) overestimates by 10%. This is the lowest value plotted at 0.895 in Figure 7. This overestimation can be overcome by selecting the reduction factor below 0.90. Since the biasness of the nominal shear resistance increases

slightly with the increasing number of ribs, two sets of inherent safety levels are provided for 4-rib shear connectors and 10-rib shear connectors. The differences in the safety levels achieved in two sets are quite negligible. It is recommended that the proposed design strength equation, Equation (5), should be applied for the design of Y-type shear connectors with more than 4 ribs. The reduction factor between the tabulated values can be selected by a linear interpolation.

**Table 23.** Shear resistance reduction factors and safety index values.

| $\phi$ | Safety Index, $\beta$ (4-Rib Case) | Safety Index, $\beta$ (10-Rib Case) |
|---|---|---|
| 0.90 | 1.593 | 1.730 |
| 0.80 | 2.288 | 2.409 |
| 0.70 | 2.982 | 3.088 |
| 0.60 | 3.676 | 3.768 |
| 0.50 | 4.371 | 4.447 |

## 5. Conclusions

In this study, the design shear resistance formula for Y-type perfobond rib shear connectors was proposed with the reduction factor. The nominal shear resistance formula was improved based on the systematic sensitivity analysis and the regression fit test with 84 experimental results. The application ranges for the basic design variables were extended and the estimation errors were reduced, compared to the formula proposed in previous studies. The basic design variables adopted in the proposed design formula were (1) number of ribs and transverse rebars, (2) concrete compressive strength, (3) rebar diameter and yield strength, and (4) rib thickness, width, height, and yield strength. The yield strengths of rebar and rib plate were additionally included to extend the application practice. The probabilistic characteristics of the shear resistance were investigated, and the various reduction factors were suggested to be selected properly depending on the target safety level in the design practice. The proposed procedure may be recommended to modify or develop the design formula for shear connectors with different shapes.

**Author Contributions:** Conceptualization, methodology, writing-review, S.-H.K.; software and formal analysis, T.B., O.H.; specimen and experiment, O.H., T.B., S.H.A.S., S.Y.; writing-draft preparation, O.H., T.B. All authors have read and agreed to the published version of the manuscript.

**Funding:** This work was supported by the Korea Institute of Energy Technology Evaluation and Planning (KETEP) and the Ministry of Trade, Industry and Energy (MOTIE) of the Republic of Korea (No. 20194030202460).

**Institutional Review Board Statement:** Not applicable.

**Informed Consent Statement:** Not applicable.

**Data Availability Statement:** Not applicable.

**Conflicts of Interest:** The authors declare no conflict of interest.

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
