# Peer review of "Development of Shear Resistance Formula for the Y-Type Perfobond Rib Shear Connector Considering Probabilistic Characteristics"

_applsci, doi:10.3390/app11093877_

Round 1

Author Response

Dear Reviewer,

We sent the revised version with our responses.

Response to Reviewer 1 Comments

Authors appreciate your valuable comments. The followings are our sincere responses to your comments:

Point 1: It is not clear how the Monte Carlo simulations are adopted in the study. The authors could better explain the use of these simulations in the context of the formula calibration.

Response 1: A brief explanation is added in the line 355-359 of new manuscript.

Point 2: Line 13: in the first line of the abstract, probably “A design shear resistance…” sounds better than “The design shear resistance…”.

Response 2: As your recommendation, it is revised in the line 13 of new manuscript.

Point 3: Line 45: “many researches are developed considering…”.

Response 3: It is revised in the line 45 of new manuscript.

Point 4: Line 68-70: the two sentences must be re-phrased, and possibly jointed.

Response 4: It is revised in the line 68-70 of new manuscript.

Point 5: Line 104: “Each test specimen consisted of”.

Response 5: It is revised in the line 104 of new manuscript.

Point 6: Line 134: “…and suggested in Eurocode 4”.

Response 6: It is revised in the line 133-134 of new manuscript.

Point 7: Line 137: “peak ultimate load” is redundant.

Response 7: It is revised in the line 138 of new manuscript.

Point 8: Line 170: please revise formulas in the text S rib(= and h’(=.

Response 8: It is revised in the line 170 of new manuscript.

Point 9: Line 207: “the new proposed resistance formula consists of only…”

Response 9: It is revised in the line 208 of new manuscript.

Point 10: Line 216 and elsewhere: “regression fit analysis”.

Response 10: It is revised in the line number 16, 216, 222, 231, 251, 275, 284, and 390 of new manuscript.

Best regards,
Sang-Hyo Kim
Yonsei Univ., Seoul

Reviewer 2 Report

The paper proposes a new design shear resistance formula for Y-type perfobond rib shear connectors. The design variables considered in the proposed formula are the following: the number of ribs and transverse rebars; the concrete compressive strength; the rebar diameter and yield strength; the rib thickness, width, height, and yield strength. The nominal shear resistance formula has been improved via a sensitivity analysis and the calibration on several experimental results.

The topic is interesting and very well developed. In my opinion some minor revisions would improve the paper for publication.

  • Bibliographic references 1, 2 and 3 are dated. Although these are basic references for the following discussion, it would be advisable to integrate them with the addition of other more recent references.
  • The coefficient of variation or the standard deviation for the values presented in Tables 4 and 5 should be entered.
  • In the conclusions it would be better to eliminate the reference to FEM numerical evaluation results, as it is a topic that is not dealt with in detail in the paper.

Author Response

Dear Reviewer,

We sent the revised version with our responses.

Response to Reviewer 2 Comments

Authors appreciate your valuable comments. The followings are our sincere responses to your comments:

Point 1: Bibliographic references 1, 2 and 3 are dated. Although, these are basic references for the following discussion, it would be advisable to integrate them with the addition of other more recent references.

Response 1: As your recommendation, references 1, 2 and 3 are changed with more recent references.

Reference 1 – Viest, I.M. Investigation of stud shear connectors for composite concrete and steel T-beams. Journal Proceedings 1956, 52(4), 875-892.

New Reference 1 - Wang, J.; Qi, J.; Tong, T.; Xu, Q.; Xiu, H. Static behavior of large stud shear connectors in steel-UHPC composite structures. Engineering Structures 2019, 178, 534-542.

Reference 2 – Viest, I.M. Full-scale tests of channel shear connectors and composite T-beams. University of Illinois at Urbana Champaign, College of Engineering. Engineering Experiment Station: Urbana, Il, USA, 1951.

New reference 2 – Shariati, M.; Sulong, N. R.; Shariati, A.; Kueh, A. B. H. Comparative performance of channel and angle shear connectors in high strength concrete composites: An experimental study. Construction and Building Materials 2016, 120, 382-392.

Reference 3 – Leonhardt, F.; Andrä, W.; Andrä, H.P.; Harre, W. New advantageous shear connection for composite structures with high fatigue strength. Beton Stahlbetonbau 1987, 82, 325-331.

New reference 3 – Zhang, J.; Hu, X.; Kou, L.; Zhang, B.; Jiang, Y.; Yu, H. Experimental study of the short-term and long-term behavior of perfobond connectors. Journal of Constructional Steel Research 2018, 150, 462-474.

Point 2: The coefficient of variation or the standard deviation for the values presented in Table 4 and 5 should be entered.

Response 2: Because the number of test data per each specimen group is too small only 3, the standard deviation or COV is not calculated.

Point 3: In the conclusions it would be better to eliminate the reference to FEM numerical evaluation results, as it is a topic that is not dealt with in detail in the paper.

Response 3: The part related to FEA results is eliminated from the conclusions.

Best regards,
Sang-Hyo Kim
Yonsei Univ., Seoul
